# Helping Others Results in Helping Yourself: How Well-Being Is Shaped by Agreeableness and Perceived Team Cohesion

**DOI:** 10.3390/bs13020150

**Published:** 2023-02-09

**Authors:** Abira Reizer, Tal Harel, Uzi Ben-Shalom

**Affiliations:** 1Department of Psychology and Behavioral Sciences, Ariel University, Ariel 44837, Israel; 2Department of Sociology and Anthropology, Ariel University, Ariel 44837, Israel

**Keywords:** well-being, agreeableness, team cohesion, leader support

## Abstract

This longitudinal research explores team cohesion as a potential mediator explaining the associations between agreeableness as a personality trait and well-being. Additionally, the study examines whether the leader offering support moderates the mediating role of perceived group cohesion. The sample consists of male military personnel (*N* = 648) from six different units. The longitudinal design examined two time points, T1 and T2, during the soldiers’ training period. The PROCESS macro for SPSS was utilized to determine the significance of the moderator and the mediation effect. The results indicate that an agreeable personality and team cohesion at T1 predicted increased well-being at T2 (two months later). In addition, the moderated mediation hypothesis was significant, indicating that when leaders offered more support, the indirect link between an agreeable personality and well-being via team cohesion was stronger than when they offered less. The findings suggest that an agreeable personality and leader support are important in the organizational domain, and these variables promote well-being. By understanding the contribution of both external and internal sources of support to soldiers’ well-being, intervention plans can be developed to lessen the stressors of their mental well-being and help them reach their maximum potential.

## 1. Introduction

The 21st century is characterized by stress, uncertainty, volatility, and constant changes, all of which challenge the personal well-being of employees [1,2]. There is thus a constantly growing amount of research that focuses on individuals’ well-being and maps the predictors that have the strongest association with well-being [3,4]. One of the stable personality traits that plays an instrumental role in predicting well-being is agreeableness [5,6].

Agreeableness is a personality trait that is typically characterized by being nice, sociable, cooperative, compassionate, empathic, and sensitive to others [7,8]. As agreeableness is a primary dimension of interpersonal behavior and social functioning [9], it positively corresponds with prosocial behaviors in general and at the workplace in particular [10].

Agreeableness has been found to be a solid predictor of well-being, happiness, and positive emotions [8,11,12]. A recent comprehensive review summarizing the 142 meta-analyses conducted to date (involving over 1.9 million participants) on the impacts of agreeableness argued that agreeableness and well-being have a medium-level effect on each other [13]. However, the authors’ conclusions were based primarily on the impact of agreeableness in the general population, which may mask more specific effects within subpopulations. They emphasized the need to reexamine the effects of agreeableness on well-being in different subpopulations. This call is in line with a recently published meta-analysis [14] indicating that there are relatively small positive effects of agreeableness on well-being at the workplace compared to stronger effects reported in non-work settings. Therefore, the next line of research needs to map the boundary conditions and the mediators in order to draw inferences from agreeableness and well-being, as both mediators and moderators remain undetected and unconsidered in the associations between agreeableness and well-being [13] (p. 268).

The current study expands the recent literature and has four main goals. The first is to examine the relationship between agreeableness and well-being in a military setting during civilians’ transition to the military organization. This transition, as well as the requirement to meet various training standards, may result in a variety of physiological and psychological symptoms in soldiers, including depression, anxiety, or physical illnesses [15,16,17]. As a result, recruits are likely to drop out [18]. Indeed, studies have shown that soldiers in the US Army who reported depression, stress, or low mental resilience were less likely to complete their training period [19]. It is common for military service to be physically and psychologically demanding, involving sensory overload, a continuously changing environment, relocations, and deployments [20,21]. Candidates should be psychologically healthy to cope with the extreme conditions experienced by military personnel. According to Cigrang, Todd, and Carbone [22], poor mental health was a significant predictor of attrition in the US within the first six months of enlistment, and the Canadian Forces also found that this factor predicted basic training attrition [23]. By examining agreeableness and well-being, the current work addresses the need to expand the current understanding of the impact of agreeableness on well-being in different subgroups. 

To advance our understanding of why and how agreeableness predicts well-being, the current work maps the potential mediators and moderators of this association. The second goal is therefore to examine whether perceived team cohesion mediates the associations between agreeableness and well-being. Team cohesion serves as an important resource and as significant social capital [24,25] that can impact employee well-being in general [26] and in military organizations in particular [27]. Since personality traits are linked to social interactions [28], they are likely to have a significant contribution on interpersonal dynamics such as team cohesion. Indeed, there is stable support for the notion that agreeable individuals are good team players [13]. Agreeable individuals are sociable and tend to invest in relationship-building and contribute to team and group development [13]; therefore, we suggest that agreeableness is related to increased well-being through the mediating role of team cohesion. 

The third goal is to examine the moderating role of leaders’ support based on the theoretical lens of self-determination theory (SDT) [29]. According to SDT, leaders’ environmental support can increase employees’ well-being. However, if the employees do not receive support, they may focus more on their self-interest and personal costs and experience greater conflicts and lower well-being. As agreeable individuals may be predisposed and environmentally influenced to be more focused on maintaining positive relationships [14], support from their leader can be very significant for them. We assume that a lack of leader support would impair the ability of an agreeable individual to invest in building stable relationships with their team members during their transition to the military and eventually impair their perceived team cohesion and well-being. 

Finally, there is very little scientific understanding of how an agreeable personality affects well-being over time, and recent additions to the general knowledge consist of cross-sectional correlations at the individual level [13]. The final goal, therefore, is to conduct a longitudinal examination of the contribution of agreeableness to well-being using a large sample longitudinal design. Figure 1 presents the study model.

### 1.1. Agreeable Personality and Well-Being

Individual differences in altruism, compassion, the ability to build positive relationships with others, help, and cooperativeness have been the subject of social interest for a long time. More recently, this pattern of behavior has been organized under the label of “agreeableness” in the Big Five model [30]. Agreeable individuals tend to be more sympathetic, considerate, friendly, cooperative, and trusting, incite liking in others, maintain positive relations, and minimize interpersonal conflict [13,31,32]. In addition, they are more likely to show active concern for the well-being of others and develop a stable social base in the organization [12,33]. 

Well-being has been defined as “feeling hopeful, happy, and good about oneself, as well as energetic and connected to others” [34] (p. 68). In this work, the definition of well-being follows Veit and Ware’s [35] conceptualization that referred to the global scale of psychological well-being as characterized by general positive affect, feelings of cheerfulness, feeling loved and wanted, and one’s satisfaction with their current positive emotional ties with other people. Psychological well-being represents the optimal psychological health of the employee [36] and reflects the theoretical conceptualization of emotional well-being [37]. Hence, it can be broadly treated as a component of mental health [38]. Indeed, several meta-analyses have suggested that agreeableness is positively associated with well-being in the general population [13,39,40,41] and in the workplace [42]. Specifically, agreeableness was positively related to both life outcomes and job satisfaction [43] and negatively associated with job burnout dimensions among employees [44]. There are some indicators of the role of agreeableness in the military setting as well. For example, a study conducted among officers in the Canadian Forces highlights the significant contribution of personality type as a predictor of soldiers’ well-being [45].

There are different reasons for the association between agreeableness and well-being. First, agreeable people tend to accept both individuals (others and themselves) as well as various circumstances, contributing to their well-being. In addition, agreeable individuals have a better sense of coherence and more positive coping strategies with challenges, such as drawing on social support, which further supports their well-being [13]. Finally, as agreeableness is considered a prosocial personality trait [5,14], givers experienced increased well-being due to their prosocial tendency (for recent meta-analyses, see [14,46,47].

Research has revealed that agreeableness is consistently related to well-being, yet the association has only been examined among limited occupations [43] based on cross-sectional data [13], and only a few studies have been conducted in a military setting. In order to maximize the external validity of the current knowledge base, the current study examines whether agreeableness is associated with well-being over time in a critical period during citizens’ transition to a military setting. Thus, we hypothesized the following: 

**H1.** 
*Agreeableness at T1 (at the beginning of the basic training) will be positively related to well-being at T2 (two months later).*


### 1.2. Agreeableness and Perceived Team Cohesion

Team members play an important role in supporting an individual’s needs and enhancing their mental well-being [48,49]. Team cohesion is one of the central variables that configure the social environment [50]. Within the group dynamics field, cohesion is commonly understood as “a dynamic process which is reflected in the tendency for a group to stick together and remain united in the pursuit of its instrumental objectives and/or for the satisfaction of member affective needs” [51] (p. 213), while perceived team cohesion describes the sense of friendship, caring for others, and closeness between group members [52].

Team cohesion is a valuable characteristic of effective military organizations, and the high-risk nature of the army values team cohesion [53]. It has been shown that team cohesion affects both performance and mental well-being [26]. Soldiers undergo intensive training in a military setting, ensuring that they will perform as expected and stay alive. Therefore, the ability to maintain team cohesion is particularly crucial in such groups [26]. According to Ben-Shalom, Lehrer, and Ben-Ari [27], cohesion affects the functioning of soldiers in combat roles. For people in high-risk occupations, such as firefighters, police, and the military, team cohesion and belongingness are integral parts of occupational identity and vital to survival [53]. Studies on the Israel Defense Forces (IDF) have also emphasized the importance of team cohesion to soldiers’ performance. Tziner and Vardi [54] showed that group cohesiveness and leadership style affect the performance effectiveness of Israeli tank crews. According to a recent study conducted in the IDF [55], group bonding interacts with leadership differentiation, enhancing group cohesion and thereby increasing group effectiveness.

Agreeableness refers to the motivation to maintain smooth interpersonal relationships [5]. This tendency among agreeable individuals to form positive relationships with others involves genetic predispositions [13] supported by active brain regions underlying reactive social behavior [56]. Consistent with this view, individual differences in agreeableness have been associated with building, maintaining, and forming positive relationships in work and non-work domains with peers, family, and spouses [13]. These positive relationships act as a source of mutual satisfaction and support when needed [5,13].

Agreeable individuals are also successful as active team players, and this trait is a predictor of team processes. Agreeable individuals can coordinate goals to cooperate effectively, accomplish team goals, enact effective conflict resolution strategies [13,57], and encourage social harmony, cooperation, and reduced competition within the group [58]. In addition, agreeable people have better reputations and are more trustworthy in social groups [59].

All these aspects support the notion that agreeableness is positively associated with team cohesion. Specifically, the individual rating of agreeableness influences the individual ratings of team cohesion beyond the influence of the actual team’s characteristics [60].

In the current work, we examine the relationship between an agreeable personality and perceived team cohesion among male military personnel in the IDF during their transition to the military. As compulsory basic military training is characterized by being both challenging and stressful [61], the contribution of agreeableness to team cohesion can be an important factor in this period. Therefore, we hypothesized the following:

**H2.** 
*Agreeableness will be positively correlated to perceived team cohesion at T1.*


### 1.3. The Mediating Role of Perceived Team Cohesion in the Relationship between Agreeableness and Well-Being

Research has shown that team cohesion is positively linked to mental well-being in the workplace [62,63]. Additionally, it is related to various physical and psychological outcomes among soldiers [64]. For example, a positive relationship was found between unit cohesion and mental well-being and growth from trauma among soldiers in the US military [65]. In a longitudinal study, increased team cohesion during the training period reduced soldiers’ psychological distress and sleeping problems and increased their sense of personal resilience and physical performance [66]. Various studies on the US military have suggested that team cohesion and a sense of belongingness were significant protective factors against suicidal ideation [67]. Team cohesion in military training has been shown to increase stress resilience [68] and decrease distress among soldiers during military training [69] and during deployment [70]. Finally, social support from peers has a favorable effect and leads to better physical and psychological health among soldiers [71].

Agreeableness engenders socially oriented tendencies and teamwork qualities [13] and is positively related to perceived team cohesion [60]. As team cohesion increases well-being, we suggest that team cohesion will mediate the relationship between agreeableness and well-being. Previous work has already acknowledged the mediating role of perceived cohesion. Bosselut et al. [72] suggested in their cross-sectional study that agreeableness would predict student engagement via the mediational role of perceived cohesion in the university classroom and called for longitudinal studies to examine whether personality and cohesion are causally related. In this study, we aim to expand the current literature by pursuing a possible linking mechanism between agreeableness and well-being in male military personnel. Therefore, we propose that the harmonious natures of agreeable people lead to higher levels of perceived team cohesion among employees, which eventually translates into better well-being over time.

**H3.** 
*Perceived team cohesion will mediate the association between agreeableness and well-being at T2.*


### 1.4. The Moderating Role of Leader Support in the Indirect Link between Agreeableness and Well-Being via Perceived Team Cohesion

Research suggests that leaders’ support plays a significant role in employees’ well-being [73]. This support is reflected in the leader’s expressions of care and concern toward the challenges faced by their subordinates [74]. Additionally, it generates motivation that improves the quality and efficiency of the work and gives employees a sense of success [75]. Research has shown that employees whose leaders provide emotional support are more likely to obtain available psychological resources [76,77].

SDT [78] provides a theoretical framework for the role of leader support. According to SDT, an individual needs to fulfill three basic needs: autonomy (experience of behavior as owned, volitional, and reflectively self-endorsed, rather than controlled), competence (experience of ability in achieving desired outcomes), and relatedness (experience of warm, caring, and mutually supportive connections with others). The satisfaction of these basic psychological needs is necessary for full functioning and organismic wellness, both in non-work and work environments, and promotes psychological, social, and physical health [78,79]. These needs can be supported by the employee’s direct leader. Specifically, when the leader supports these three needs, the employee experiences several positive outcomes, including well-being, psychological health, social wellness, and work-related functioning [80,81]. For example, police officers were more autonomously motivated when their leaders supported their need for autonomy, while health professionals who received support from their leaders reported better work satisfaction and psychological health [82,83]. Additional studies demonstrated that leadership support predicted occupational and psychological outcomes, such as job satisfaction and health symptoms in the US military [84] and less occupational stress and strain in the UK Royal Navy [85].

Leaders also have a significant impact on the affective reactions of team members [86]. One recent study has shown that supportive leaders may set the tone and norms within a unit and contribute to cultivating a cohesive culture among coworkers [87]. In response to their leader’s support, team members may exert more effort at work and invest more in teamwork. Empirical studies provide evidence that employees who have better relationships with their leaders perform better, are more committed, and engage in more helpful behavior [88,89]. Furthermore, leaders who demonstrate understanding and coach their employees are more likely to foster a cooperative work environment where employees feel appreciated [88,90].

Despite evidence for the benefits of leader support concerning various outcomes on the individual and teamwork levels, little information in the literature focuses on its moderating role in agreeableness and well-being literature. Recently, Liao et al. [14] suggested that an SDT framework should be incorporated into research on prosocial personality motivation and well-being. Specifically, they argued that autonomous rather than obligatory prosocial motivation would moderate relationships between prosocial motivation and well-being. We develop this line of reasoning by suggesting that leader support is crucial for agreeable individuals to provide a nurturing environment for others by facilitating teamwork and team cohesion and, eventually, to increase their own well-being. Our research thus offers a comprehensive conceptualization of the need for leaders’ support regarding the three basic psychological needs (autonomy, competence, and relatedness) based on SDT. We suggest that leaders’ support of these needs may change the extent to which agreeableness predicts perceived team cohesion. Therefore, we hypothesized the following:

**H4.** 
*Leader support at T1 moderates the indirect relationship between agreeableness at T1 and well-being at T2 via perceived team cohesion at T1, such that the indirect effect will be stronger when leader support is high than when it is low.*


## 2. Materials and Methods

### 2.1. Participants

The sample consisted of 648 male soldiers, aged between 18 and 25, from six Israeli Ground Forces Command battalions. The study was performed at two different time points during their training. At the first time point (T1) (the beginning of the basic training), 1374 participants completed the survey. Two months later, at the second time point (T2) (the end of basic training), 648 participants completed it. Between T1 and T2, 726 participants dropped out (a rate of 52.8%). 

### 2.2. Instruments

*Demographic data survey*. A brief survey was used to collect the following demographic data from the participants: (a) year of birth, (b) place of birth, (c) religion, (d) gender, (e) date of enlistment, (f) current stage of training, and (g) whether they chose to serve in their current unit.

*Well-being.* Psychological well-being was assessed using an abbreviated version of the Mental Health Inventory (MHI) [35] translated into Hebrew by Florian and Drury [91]. This scale’s development was intended to extend the definition of mental health by including characteristics of well-being (e.g., feeling cheerful, interest in and enjoyment of life) [35] (p. 730). The psychological well-being subscale includes 14 items measuring one’s psychological well-being (general positive affect and satisfaction with one’s emotional ties) and includes items that assess positive aspects of well-being, such as “I woke up expecting an interesting day.” Soldiers rated their agreement with the items using a 6-point Likert scale ranging from “None of the time” to “All the time.” These had good reliability results, with Cronbach’s alpha coefficients of α = 0.97 (T1) and α = 0.96 (T2). Based on a summation of the employees’ evaluations, a psychological well-being score of 14 to 84 was calculated, where a higher score indicates a better state of well-being. The MHI was developed and psychometrically tested in English-speaking countries but has since been translated into many languages [92,93], including Hebrew. An analysis of 605 men and women in Israel conducted by Florian and Drury [91] confirmed the construct’s validity and the external validity of the Hebrew questionnaire.

*Agreeableness.* We used the Mini-IPIP (International Personality Item Pool) scale developed by Donnellan et al. [94], which contains 20 items. Each Big Five trait was measured using four items in this shortened version. For example, one item of the agreeableness dimension was “I sympathize with others’ feelings.” The participants were asked to indicate the extent to which each item characterized them on a 5-point Likert scale, ranging from “Not at all” to “Very much.” This study focused on the agreeable personality trait, which showed good reliability (α = 0.77).

*Perceived team cohesion*. Based on the scale of Podsakoff and MacKenzie [95], this scale includes three items, such as “My teammates know they can trust each other.” The participants ranked each item on a 7-point Likert scale, ranging from “Strongly disagree” to “Strongly agree.” A previous study has used this abbreviated scale in the military context [96]. In the current study, the three-item scale produced a good reliability coefficient (α = 0.89).

*Leadership support.* The participants responded to a 12-item measure [82], the items of which were adapted to examine leadership support in the IDF and included, for example, “I feel that my leader understands me.” The respondents were asked to rate the items on a 7-point Likert scale ranging from “Do not agree at all” to “Completely agree.” The scale yielded good reliability coefficients for all 12 items (α = 0.80). 

### 2.3. Study Design and Procedure

The participants were recruited through collaboration with the IDF, which was interested in longitudinal research on employees’ well-being and motivation. The participants were men who served in one of the six units of the Israeli Ground Forces Command. The study was approved by the ethics committee of the Faculty of Social Sciences and Humanities at Ariel University and the IDF.

Data were collected from 2017 to 2018 in two waves; T1 was at the beginning of the participants’ basic training (3–4 weeks from the date of enlistment), and T2 was at the end of their basic training (two months later).

Prior to enrolling in the study, the participants reviewed the informed consent form and acknowledged their understanding and willingness to take part in the study voluntarily. Participants were allowed to drop out of the study at any time. In accordance with the requirements of the research ethics committee, participants were given 48 h at home for consultation and consideration before signing their informed consent form.

The questionnaires were handed over by an external experimenter from outside the military. When the questionnaires were handed over, no other people except the experimenter and the participants were in the room. The data were collected using the paper and pencil technique to ensure the anonymity and confidentiality of the respondents. Participants who agreed to participate in the study were required to fill out five questionnaires: (a) a demographic survey, (b) the well-being scale, (c) the big five personality trait scale, (d) the perceived team cohesion scale, and (e) a leadership support survey. The approximate time required for participation was 20 min at each time point.

## 3. Results

### 3.1. Preliminary Analysis

Before testing our hypotheses, we conducted a confirmatory factor analysis (CFA) using structural equation modeling (SEM) to assess the convergent and discriminant validity of the core variables. The model included four latent variables: agreeableness, well-being, perceived team cohesion, and leader support. The inter-correlations between the latent factors and the factor loadings with the items were included in the model. The results showed high factor loadings (higher than 0.40) for each measurement item on the factor it was designed to estimate, except for item nine on the leader support scale, which yielded an average factor loading (0.36). The model fitted the data well: χ2(388) = 1273.558, *p* < 0.001, CFI = 0.92, TLI = 0.91, and RMSEA = 0.06.

Table 1 presents the means, standard deviations, and intercorrelations of the study variables. As can be seen in Table 1, agreeableness at T1 was positively related to well-being at T1 and T2 and perceived team cohesion at T1. Additionally, leader support at T1 was positively associated with perceived team cohesion at T1. These results support H1 and H2.

### 3.2. Moderated Mediation Model

In order to test the moderated mediation model posited in H3 and H4, we used the PROCESS macro for SPSS (Model 7), and 5000 bootstrap samples were adopted to examine the moderated mediation effects [97]. The results revealed that the direct effect of agreeableness personality at T1 on well-being at T2 was significant (B = 0.1255, SE = 0.0478, 95% CI [0.0317–0.2193]). The interaction between agreeableness at T1 and leader support at T1 was also significant (B = 0.1177, SE = 0.0436, 95% CI [0.0320–0.2034]). Additionally, leader support at T1 significantly moderated the indirect effect of agreeableness at T1 on well-being at T2 via perceived team cohesion at T1 (index of moderated mediation effect: B = 0.0128, SE = 0.0078, 95% CI [0.0009–0.0303]). This means that at higher levels of leader support (*Mean level* and +1 *SD*), the indirect effect of agreeableness personality at T1 and well-being at T2 through perceived team cohesion at T1 is significantly stronger than at lower levels of leader support (B = 0.0134, SE = 0.0077, 95% CI [0.0011–0.0309]; B = 0.0261, SE = 0.0133, 95% CI [0.0044–0.0561], respectively). The indirect effect was no longer significant at the lower level of leader support (−1 SD). This model explained 16.13% of the variance in the soldiers’ well-being at T2 (*p* < 0.01). Table 2 and Table 3 present the results, which generally support H3 and H4.

## 4. Discussion

This study sought to expand understanding of the mechanisms related to well-being by investigating the mediating role of perceived team cohesion in the association between an agreeable personality and well-being over time, as well as the moderating role of leader support in this indirect link. The causal processes relating personality and well-being remain unclear, so this study used a longitudinal model to examine the relationships between agreeableness, perceived team cohesion, and well-being at two time points during employees’ recruitment process and onboarding phase. Additionally, we asked whether leaders’ support moderates the indirect relationship between agreeableness and well-being via perceived team cohesion.

The findings showed positive associations between agreeableness and well-being over time (at both T1 and T2). Moreover, agreeableness was positively related to perceived team cohesion. These results support H1 and H2. Agreeableness (along with extraversion) is considered the personality factor responsible for satisfying relationships and the quantity and quality of interpersonal relationships [12]. In addition, according to Graziano and Eisenberg [5], agreeableness is a core dispositional trait contributing to prosocial behavior, which includes a broad range of actions that are intended to benefit others and are valued by society [5,98,99]. Individuals’ willingness to sacrifice was positively associated with their own personal and their relationships’ well-being [47]. These findings correspond to the notion that agreeableness as a prosocial tendency can benefit not only others but also oneself [14].

In response to calls for researchers to explore the extent, boundary conditions, and mediating process of agreeableness and well-being [13], our findings indicate that team cohesion mediates the association between agreeableness and well-being, thus supporting H3. A team context such as the army requires members to interact and rely on each other for information and support, and cooperative behavior facilitates effective coordination between diverse ideas and contributions within a team unit [100]. Indeed, studies have found that agreeableness is positively related to well-being and agreeableness within a team, and to team effectiveness and performance [101]. In addition, caring individuals tend to allocate scarce personal resources to others [102]. Furthermore, we found that this tendency not only improves teamwork (cohesion) but can also produce favorable personal outcomes, such as increased personal well-being. These results correspond with a recent meta-analysis, in which it was argued that relational investment (i.e., positive relationships) and teamworking (i.e., coordination with others) are solid characteristics of agreeableness and that these constructs are considered “as a firm foundation of knowledge and as a scaffold for future research and theory” [13] (p. 243). 

In addition, the research findings echo the existing theory that classifies prosocial tendencies as primarily altruistic or primarily selfish [98,103]. Prosocial acts can be a result of selfish or egocentric motives to increase benefits for the self [104], such as increased reputation [105], boosts to one’s self-esteem and mood [106], and expectations of future help from others [107]. Our findings indicated that the support given by significant others is a moderator that impacts the tendency of prosocial agreeable individuals to increase group cohesion and eventually create more positive personal outcomes. Therefore, it incorporates benefits both to others and to the self.

Finally, in line with H4, the moderated mediation model was significant, indicating that leaders’ support of soldiers’ needs significantly moderated the indirect effect of agreeableness on well-being via perceived team cohesion. This means that at higher levels of leader support, the indirect effect of agreeableness and well-being through perceived team cohesion was significantly stronger than at lower levels. While extensive research has highlighted the robustness of the relationship between agreeableness and well-being, potential moderators are still under investigation. However, the current findings support the notion that SDT [29,108,109] can serve as a theoretical framework for mapping associations between agreeableness and well-being.

### 4.1. Theoretical and Practical Contributions

There is strong interest in understanding how agreeableness can be beneficial both for people’s relationships and for their own well-being, and this work expands on Wilmot and Ones’ [13] argument that agreeableness affects teamwork by indicating that teamwork cohesion serves as a significant moderator. It also supports the notion that interacting with others and creating positive, supportive relationships are important components of how people feel at work [110].

Agreeable individuals function better under supportive leaders. Consistent with this view, Judge and Cable [111] found that agreeable job seekers were more attracted to organizations with supportive and team-oriented cultures than less agreeable individuals were. Therefore, the next theoretical line of research may investigate the contribution of the interpersonal nature of environments [112] as a potential boundary for agreeable individuals.

At a more practical level, there is a growing tendency to develop and promote interventions that support the health and well-being of individuals e.g., [2,3]. As agreeableness serves as a predominant factor for well-being, future interventions should take it into consideration while conducting further research. In military organizations, being able to predict psychological well-being based on agreeableness characteristics has important implications, including reducing selection costs and attrition and improving personnel morale [113]. In addition, building and developing a cohesive environment in the military setting may be an effective strategy to promote soldiers’ well-being.

While the underlying assumption is that agreeableness can serve as a catalyst for psychological well-being, not all candidates are characterized by this trait when they enter a job. Blackie and colleagues [114] reviewed empirical evidence for interventions that instruct individuals to enact behaviors of certain personality traits that may be effective for enhancing well-being, positive affect, work performance, and creative thinking [115,116,117]. A similar approach and instructing individuals to behave agreeably (e.g., being cooperative and kind) could therefore be an effective strategy for promoting this quality rather than taking it as a given personality trait.

Our findings suggest that leaders’ support of the three basic needs (e.g., autonomy, competence, and relatedness) can improve agreeableness functioning and positive outcomes. According to SDT, managers contribute significantly to employees’, as well as soldiers’, basic psychological needs [78]. Leaders can support the need for autonomy by asking and acknowledging employees’ perspectives and feelings before taking action, supporting employees’ decision-making processes and proposals, providing meaningful justifications, and minimizing force and intimidation. To foster competence, leaders can express their authentic belief in employees’ capability to accomplish their tasks, recognize obstacles, and provide feedback with a non-judgmental approach. To support relatedness, leaders can show unconditional positive interest, remain empathetic to soldiers’ concerns, and create a warm and accepting interpersonal environment even when they do not meet their expectations. In essence, supporting these basic needs means that leaders care about their subordinates, actively engage with them, and adopt an other-centered perspective in their interactions with them [81]. Our findings suggest that this approach can have the same impact in stressful military environments.

### 4.2. Limitations and Future Research

Several limitations of the current work need to be acknowledged. First, the application of the findings is limited to specific circumstances and to a country-specific military culture. Based on a person-environment fit framework [118,119,120,121] and, in particular, a situational congruence model positing that individuals will perform better in environments that are congruent with their personalities e.g., [121], agreeableness is the personality dimension that may best fit the interpersonal demands associated with teamwork processes in this stressful environment. Future work may generalize the current findings in a work setting with less stressful demands. Therefore, different contexts or levels of analysis, occupations, teams, and cultures may have important implications. Second, this study measured agreeableness as a unidimensional construct, similar to previous work e.g., [13]; however, a growing body of evidence shows the usefulness of a multidimensional measurement of the agreeableness construct [8,39]. Third, while team cohesion is a dynamic process, our study design only allowed us to capture a fixed picture of the construct. Data collection took place at the beginning and end of the training period; however, changes in cohesion can occur throughout a study period. Studies with repeated measures for long-term periods would be better suited to validate our model.

Future studies can employ a wider range of occupations and periods to maximize the model’s external validity, as agreeable individuals are more successful in adjusting to different novel environments and institutions [13,72]. In addition, future work can expand on the hindrances and costs of being agreeable in the workplace, especially when the trait has zero or negative relations with extrinsic career success, such as productivity, academic and training success, or salary [13]. Furthermore, future research may include female or mixed-gender teams alongside male teams in order to gain a broader perspective on agreeableness, team cohesion, and well-being.

In summation, as McAdams argues, “agreeable people are more than nice. Agreeableness incorporates expressive qualities of love and empathy, friendliness, cooperation and care … [and] includes such concepts as altruism, affection and many of the most admirably humane aspects of the human personality” [122] (pp. 89–90). Following this argument, the current findings shed light on the boundary conditions under which agreeableness is linked with intrapersonal and interpersonal benefits and consequences. The results indicate that individuals with higher levels of agreeableness experience greater well-being in the military setting due to an increased perception of team cohesion, and this positive mechanism is dependent upon external leader support. These findings may inform individuals as well as organizations on how to leverage well-being to their best interests.

## Figures and Tables

**Figure 1 behavsci-13-00150-f001:**
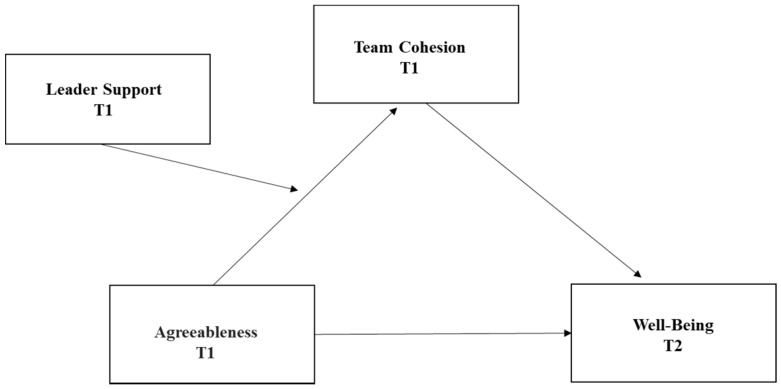
Study model.

**Table 1 behavsci-13-00150-t001:** Descriptive statistics and correlations among the research variables (*N* = 648).

	M	SD	1	2	3	4	5
Well-being T1	4.13	1.18	-				
Well-being T2	3.98	1.15	0.45 ***	-			
Agreeableness T1	5.58	0.95	0.21 ***	0.11 **	-		
Team cohesion T1	5.03	1.22	0.20 ***	0.12 ***	0.10 *	-	
Leader support T1	3.25	1.13	0.24 ***	0.07	0.08 *	0.10 **	-

*Note*. * *p* < 0.05, ***p* < 0.01, *** *p* < 0.001.

**Table 2 behavsci-13-00150-t002:** Moderated mediation model.

Mediator Variable (Team Cohesion T1)						
	*B*	SE	*T*	*p*	LLCI	ULCI
Constant	6.2624	0.8844	7.0811	0.0000	4.5258	7.9990
Agreeableness T1	−0.2803	0.1549	−1.8097	0.0708	−0.5844	0.0238
Leader support T1	−0.5605	0.2515	−2.2285	0.0262	−1.0544	−0.666
Agreeableness T1 × Leader support T1	0.1177	0.436	2.6976	0.0072	0.0320	0.2034
Conditional effects of the predictor at the values of the moderator (leader support T1)						
Leader support T1	Effect	SE	*T*	*p*	LLCI	ULCI
2.2857	−0.0112	0.0688	−0.1634	0.8703	−0.1463	0.1238
3.4286	0.1233	0.0506	2.4343	0.0152	0.0238	0.2227
4.4286	0.2410	0.0689	3.4971	0.0005	0.1057	0.3763
Dependent variable model (well-being T2)						
Constant	2.7360	0.3143	8.7058	0.0000	2.1189	3.3531
Agreeableness T1	0.1255	0.0478	2.6270	0.0088	0.0317	0.2193
Team cohesion T1	0.1085	0.0368	2.9467	0.0033	0.0362	0.1808

*Note*. Values for leader support T1 are the mean and +/−1 SD from the mean.

**Table 3 behavsci-13-00150-t003:** Direct and conditional indirect effects.

Direct Effect of Agreeableness T1 on Well-Being T2					
Effect	SE	*t*	*p*	LLCI	ULCI
0.1255	0.0478	2.6270	0.0088	0.0317	0.2193
Conditional indirect effects					
Mediator	Leader support T1	Effect	SE	LLCI	ULCI
Team cohesion T1	2.2857	−0.0012	0.0084	−0.0202	0.0143
Team cohesion T1	3.4286	0.0134	0.0077	0.0011	0.0309
Team cohesion T1	4.4286	0.0261	0.0133	0.0044	0.0561
Index of moderated mediation					
Mediator	Index	SE	LLCI	ULCI	
Team cohesion T1	0.0128	0.0078	0.0009	0.0303	

*Note*. Values for leader support T1 are the mean and +/−1 SD from the mean.

## Data Availability

The data presented in this study are available on request from the corresponding author.

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
