# Peer review of "Helping Others Results in Helping Yourself: How Well-Being Is Shaped by Agreeableness and Perceived Team Cohesion"

_behavsci, 2023, doi:10.3390/bs13020150_

Round 1

Reviewer 1 Report

Dear Authors,

The connection between agreeableness and well-being is not new and the research model is simple, however, expanding the understanding of the impact of agreeableness on well-being in the subgroup of army solders and in a longitudinal study is quite unique and contributing the the academic research. Congratulations!

Comments for review:

lines 69-72 are not consistent. for example, "Altruism, cooperation, trust, and compassion are all necessary ingredients of a harmonious and well-functioning society" why is it mentioned? How is it related to demonstrating  team cohesion as a mediator? this paragraph should be revised.

It might be useful to enrich this paper with more references regarding team cohesion in the Israeli Army (IDF) as this is the subgroup you research. 

for example:

Klang, M. (2017). The influence of work group bonding on effectiveness: Group cohesion versus transactive memory mediating role and boundary conditions of the process. [PhD dissertation, University of Haifa].

Tziner, A., & Vardi, Y. (1982). Effects of command style and group cohesiveness on the performance effectiveness of self-selected tank crews. Journal of Applied Psychology, 67(6), 769–775. https://doi.org/10.1037/0021-9010.67.6.769

No items demonstrate the agreeableness questionnaire (lines 279-284)

A reference to the demographic variables you collected should be inserted, were they of any significance (for example, religion?).

Also as all the subjects were male, in future research it is valuable to research, in the army, team cohesion of female teams or even mixed gender teams.

Author Response

Reviewer 1

The connection between agreeableness and well-being is not new and the research model is simple, however, expanding the understanding of the impact of agreeableness on well-being in the subgroup of army solders and in a longitudinal study is quite unique and contributing the the academic research. Congratulations!

 Author Response: We thank the reviewer for the supportive comment.

Comments for review:

Reviewer Comment 1: lines 69-72 are not consistent. for example, "Altruism, cooperation, trust, and compassion are all necessary ingredients of a harmonious and well-functioning society" why is it mentioned? How is it related to demonstrating team cohesion as a mediator? this paragraph should be revised.

Author Response 1: Thank you for your comment. We revised the paragraph (lines 66-77).

Reviewer Comment 2: It might be useful to enrich this paper with more references regarding team cohesion in the Israeli Army (IDF) as this is the subgroup you research. 

Author Response 2: Thank you for your valuable comment. We have incorporated these articles into the introduction section (see lines 158-163).

Reviewer Comment 3: No items demonstrate the agreeableness questionnaire (lines 279-284)

Author Response 3: Thank you for the comment. An example item has now been added in the method section (lines 295-296).

Reviewer Comment 4: A reference to the demographic variables you collected should be inserted, were they of any significance (for example, religion?).

Author Response: Thank you for the comment. The demographic variables were analyzed, but no significant differences were found.

Reviewer Comment 4: Also as all the subjects were male, in future research it is valuable to research, in the army, team cohesion of female teams or even mixed gender teams.

Author Response: Thank you for mentioning this important point. We have added this idea in the discussion section (lines 491-493).

Reviewer 2 Report

Overall, a well-conducted and presented research study.  My suggestions would be rather minor and possibly a matter of taste rather than substance.  

While agreeableness is shown to be important, you might take a few moments (sentences) to speculate on how to develop and/or promote this quality rather than taking it as a personality given.  For example, the military can't reject candidates who don't have this characteristic on entrance, but what might they do to better promote its development (the stereotypical drill sergeant, for example, is not particularly agreeable)?

Author Response

Thank you for the supportive comments.

 Reviewer Comment 1: While agreeableness is shown to be important, you might take a few moments (sentences) to speculate on how to develop and/or promote this quality rather than taking it as a personality given.  For example, the military can't reject candidates who don't have this characteristic on entrance, but what might they do to better promote its development (the stereotypical drill sergeant, for example, is not particularly agreeable)?

Author Response 1: Thank you for your valuable comment. We have added a paragraph addressing possible interventions that may develop and promote agreeable behaviours (see lines 446-453).